# Intracellular Mg$^{2+}$ protects mitochondria from oxidative stress in human keratinocytes

Keigo Fujita[1,6], Yutaka Shindo[1,2,6], Yuji Katsuta[3], Makiko Goto[3], Kohji Hotta[1] & Kotaro Oka [1,2,4,5✉]

Reactive oxygen species (ROS) are harmful for the human body, and exposure to ultraviolet irradiation triggers ROS generation. Previous studies have demonstrated that ROS decrease mitochondrial membrane potential (MMP) and that Mg$^{2+}$ protects mitochondria from oxidative stress. Therefore, we visualized the spatio-temporal dynamics of Mg$^{2+}$ in keratinocytes (a skin component) in response to H$_2$O$_2$ (a type of ROS) and found that it increased cytosolic Mg$^{2+}$ levels. H$_2$O$_2$-induced responses in both Mg$^{2+}$ and ATP were larger in keratinocytes derived from adults than in keratinocytes derived from newborns, and inhibition of mitochondrial ATP synthesis enhanced the H$_2$O$_2$-induced Mg$^{2+}$ response, indicating that a major source of Mg$^{2+}$ was dissociation from ATP. Simultaneous imaging of Mg$^{2+}$ and MMP revealed that larger Mg$^{2+}$ responses corresponded to lower decreases in MMP in response to H$_2$O$_2$. Moreover, Mg$^{2+}$ supplementation attenuated H$_2$O$_2$-induced cell death. These suggest the potential of Mg$^{2+}$ as an active ingredient to protect skin from oxidative stress.

[1] Department of Bioscience and Informatics, Faculty of Science and Technology, Keio University, Yokohama, Japan. [2] School of Frontier Engineering, Kitasato University, Sagamihara, Japan. [3] MIRAI Technology Institute, Shiseido Co. Ltd., Yokohama, Japan. [4] Waseda Research Institute for Science and Engineering, Waseda University, Tokyo, Japan. [5] Graduate Institute of Medicine, College of Medicine, Kaohsiung Medical University, Kaohsiung City, Taiwan. [6]These authors contributed equally: Keigo Fujita, Yutaka Shindo. ✉email: oka@bio.keio.ac.jp

Reactive oxygen species (ROS) are constantly produced in the human body and have harmful effects. Exposure to ultraviolet (UV) irradiation in particular is a notable trigger for ROS generation[1]. ROS generation is attributed to several factors represented by enzyme activities of the electron transport chain in mitochondria[2]. Oxidative stress induced by ROS constitutes a harmful condition because ROS have high reactivity and cause DNA damage, lipid peroxidation, and protein carbonylation. These forms of oxidative damage increase with age[3,4]. Excessive ROS generation also causes mitochondrial dysfunction. In cellular-level experiments, the addition of $H_2O_2$ (a type of ROS) decreased mitochondrial membrane potential (MMP)[5,6]. Moreover, it has been reported that ROS-induced mitochondrial dysfunction contributes to various diseases, including Alzheimer's disease, type 1 diabetes, atherosclerosis, and cancer[7]. Skin in particular is frequently exposed to ROS stress since UV irradiation from sunlight is the main generator of ROS[1]; ROS stress has been suggested to be involved in aging, inflammation, and pathogenesis of skin cancer, among other conditions[8,9]. Therefore, protecting biomolecules and mitochondria from ROS is critical for maintaining normal cellular functions, and cells express antioxidants and several mechanisms to avoid oxidative stress[10]. Skin cells, such as the keratinocytes and fibroblasts which constitute the epidermis, also have ROS clearance mechanisms[11]. In our previous study, we suggested that $Mg^{2+}$ was involved in mechanisms to help cells avoid oxidative stress[12].

$Mg^{2+}$ is the most abundant divalent cation in living cells and is related to more than 600 enzymes as a cofactor[13–15]. Recent studies have demonstrated that small changes in intracellular $Mg^{2+}$ can have large impacts on cellular events, such as cell division, maturation of neurons, and neurodegeneration[16–18]. In neurons and cancer cells, intracellular $Mg^{2+}$ is stored in the mitochondria and constitutes $Mg^{2+}$ an important factor for sustainable mitochondrial functioning[12,19]. The impact of $Mg^{2+}$ in relation to retaining MMP has been reported at the cellular and isolated mitochondrial levels[20–22]. The protective effects of $Mg^{2+}$ toward oxidative stress have also been reported in various types of cells, such as endothelial cells[23,24], bone marrow mesenchymal stem cells[25,26], and chick embryo hepatocytes[27]. On the other hand, in monocytes the environment is the determining factor in whether high levels of $Mg^{2+}$ will increase or decrease ROS levels[28]. These findings clearly indicate the important role of $Mg^{2+}$ in protecting cells from oxidative stress. However, it has not been well understood whether cells change intracellular $Mg^{2+}$ concentrations in response to oxidative stress, nor exactly how $Mg^{2+}$ protects cells from oxidative stress.

In this study, we examined changes in $Mg^{2+}$ concentration in response to ROS in keratinocytes. Keratinocytes were chosen because they are one of the most ROS-exposed cells in the human body, as ROS are generated by UV from sunlight. The dynamics of cytosolic $Mg^{2+}$ were visualized using fluorescence imaging under $H_2O_2$ stress to find that $H_2O_2$ induced an increase in cytosolic $Mg^{2+}$ concentration ($[Mg^{2+}]_{cyto}$) due to the release of $Mg^{2+}$ bound to ATP upon the $H_2O_2$-induced decrease in ATP level. The effects of $Mg^{2+}$ on mitochondrial functions were investigated by comparison to changes in MMP to find that the increased $Mg^{2+}$ attenuated the decrease in MMP. Interestingly, $Mg^{2+}$ supplementation further suppressed the decrease in MMP and $H_2O_2$-induced cell death. Our results suggest that $Mg^{2+}$ provides robustness to intracellular ATP levels under oxidative stress.

## Results

### $H_2O_2$-induced cytosolic $Mg^{2+}$ increase.
Spatio-temporal dynamics of $Mg^{2+}$ in keratinocytes derived from adults (adult keratinocytes) were visualized with an $Mg^{2+}$ selective indicator, KMG-104. It was confirmed that $H_2O_2$ did not directly react with KMG-104 (Supplementary Fig. 1a). $H_2O_2$ caused the $[Mg^{2+}]_{cyto}$ to increase in keratinocytes from 40-year-old donor (Fig. 1a–c). The average time-course of $[Mg^{2+}]_{cyto}$ increased immediately after application of 1 mM $H_2O_2$ and reached a plateau at 5 min (Fig. 1a). The spatial distribution of the changes in $[Mg^{2+}]_{cyto}$ was almost uniform within the cells, whereas the amplitude of change in each cell varied (Fig. 1c). To examine whether the $H_2O_2$-induced $Mg^{2+}$ increase constituted a common phenomenon in keratinocytes across individual differences and age, the responses were compared in five keratinocyte cell lines: three from newborns (three different 0 years old donors: newborn keratinocytes) and two from adults (40 and 57 years old donors: adult keratinocytes). There was large variability in the amplitudes of the $H_2O_2$-induced $Mg^{2+}$ response in each cell, regardless of age (Fig. 1d and Supplementary Fig. 2a). While some newborn keratinocytes increased and others decreased in $[Mg^{2+}]_{cyto}$ in response to $H_2O_2$ for all three cell lines, most of the adult keratinocytes showed increases in $[Mg^{2+}]_{cyto}$. As a result, all newborn keratinocyte lines showed no or slightly decreased $[Mg^{2+}]_{cyto}$ responses on average, whereas adult keratinocytes lines exhibited increased $[Mg^{2+}]_{cyto}$ on average (Supplementary Fig. 2b). Therefore, the data were divided into two groups, newborn and adult, and the difference in these two groups were compared. The amplitude of $H_2O_2$-induced $Mg^{2+}$ responses was significantly larger in adult keratinocytes than in newborn keratinocytes (Fig. 1e). A higher concentration of $H_2O_2$ (10 mM) elicited increases in $[Mg^{2+}]_{cyto}$, even in newborn keratinocytes, indicating that this is not a phenomenon specific to adult keratinocytes, but simply that adult keratinocytes are more sensitive to $H_2O_2$ than newborn keratinocytes (Supplementary Fig. 2c, d). The responses to 10 mM $H_2O_2$ were also larger in adult keratinocytes than in newborn keratinocytes (Supplementary Fig. 2e, f). To examine underlying mechanism of the increase in $[Mg^{2+}]_{cyto}$ and role of the $Mg^{2+}$, the following experiments were performed in adult keratinocytes from a 40-year-old donor.

### A major source of $Mg^{2+}$ is $Mg^{2+}$ dissociation from ATP in the process of ATP consumption.
To identify the $Mg^{2+}$ source that was responding to $H_2O_2$, we examined $Mg^{2+}$ entry from an extracellular medium, and fluorescence imaging was performed in the medium without $Mg^{2+}$. An $H_2O_2$-induced $Mg^{2+}$ increase was still observed in this condition (Fig. 2a), which indicates that $H_2O_2$ induced the release of $Mg^{2+}$ from an intracellular $Mg^{2+}$ source. In other cell types, mitochondria were identified as intracellular $Mg^{2+}$ storage sites and released $Mg^{2+}$ into cytosol upon a depolarization of MMP[19,29,30]. Simultaneous imaging of $[Mg^{2+}]_{cyto}$ and MMP revealed that FCCP, an uncoupler of mitochondria, rapidly decreased MMP; however, FCCP did not increase $[Mg^{2+}]_{cyto}$ in keratinocytes, but rather deceased it (Fig. 2b). This indicates that depolarization of the mitochondria did not cause $Mg^{2+}$ release from mitochondria in keratinocytes.

Next, the involvement of $Mg^{2+}$ transporters was investigated. It has been reported that $Na^+/Mg^{2+}$ exchangers, one of which is SLC41A1, mediate $Mg^{2+}$ efflux from the cells and are inhibited by quinidine[14,22,31]. Therefore, if one of the targets of $H_2O_2$ is the $Na^+/Mg^{2+}$ exchanger, which leads to the increase in $[Mg^{2+}]_{cyto}$, it is expected that quinidine also increases $[Mg^{2+}]_{cyto}$ and that prior application of quinidine abolishes the $H_2O_2$-induced increase in $[Mg^{2+}]_{cyto}$. Quinidine alone induced increase in $[Mg^{2+}]_{cyto}$ in both normal and $Mg^{2+}$-free conditions, whereas vehicle (DMSO, final concentration 0.5%) decreased $[Mg^{2+}]_{cyto}$, indicating that inhibition of $Mg^{2+}$ efflux leads to increase in $[Mg^{2+}]_{cyto}$ in keratinocytes (Supplementary Fig. 3a). The amplitude of quinidine-induced $Mg^{2+}$ increase was greater in

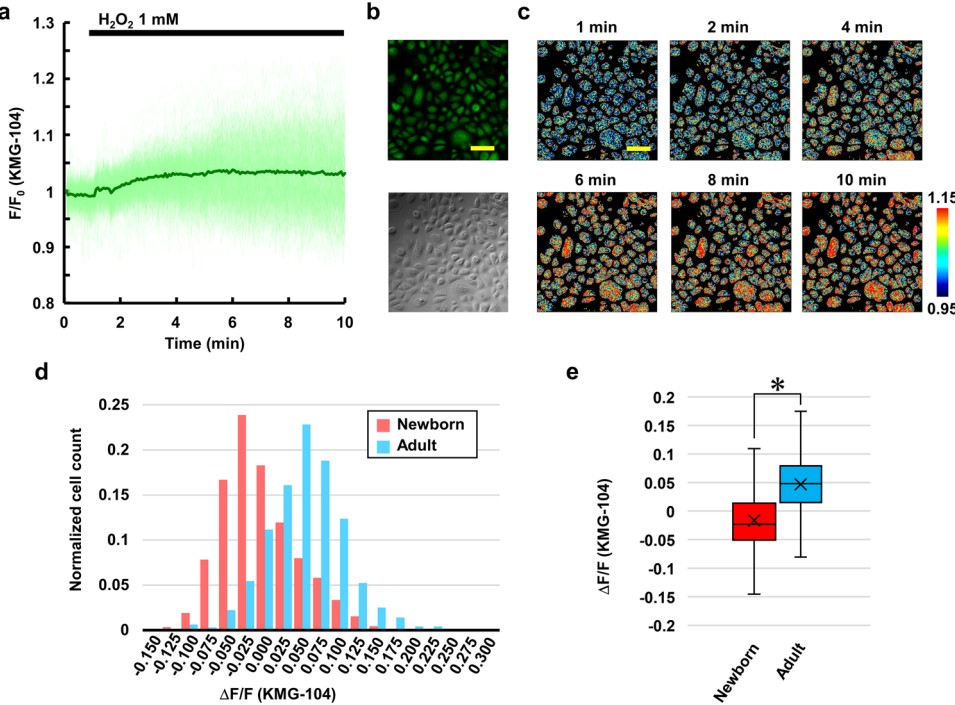

**Fig. 1 $[Mg^{2+}]_{cyto}$ increase in response to $H_2O_2$ in keratinocytes. a** Time-course of $[Mg^{2+}]_{cyto}$ in response to $H_2O_2$ (1 mM) added at 1 min in adult keratinocytes. Mean (green line) and all traces (light green lines) ($n = 511$ cells from four different experiments). **b** Fluorescence image of keratinocytes stained with KMG-104 and differential interference contrast (DIC) image. **c** Pseudo-color image of $Mg^{2+}$ dynamics ($F/F_0$) in adult keratinocytes at the indicated time points. **d** Histogram showing the distribution of $Mg^{2+}$ responses induced by $H_2O_2$ (1 mM) in newborn keratinocytes (red histogram: $n = 1888$ cells from nine different experiments, which include data from three keratinocyte cell lines; 0 years-1: $n = 594$ cells from three different experiments, 0 years-2: $n = 598$ cells from three different experiments, and 0 years-3: $n = 696$ cells from three different experiments) and adult keratinocytes (blue histogram: $n = 994$ cells from seven different dishes, which include data from two keratinocyte cell lines; 40 years: $n = 511$ cells from four different experiments, and 57 years: n = 483 cells from three different experiments). **e** Comparison of the average amplitude of $Mg^{2+}$ response shown in **d**. The amplitude was calculated as a difference between the average of $F/F_0$ before (0–1 min) and after (9–10 min) $H_2O_2$ treatment. Center line: median, x: average, box limits: quartiles, whiskers: 1.5× interquartile range. Scale bar in this figure: 100 µm. *$p < 0.05$ (Student's $t$-test, two-sided).

normal condition than $Mg^{2+}$-free condition (Supplementary Fig. 3b). These results suggest that $[Mg^{2+}]_{cyto}$ is normally balanced by $Mg^{2+}$ influx and efflux. The effect of quinidine on $H_2O_2$-induced $Mg^{2+}$ responses was also investigated. The responses were greater in the presence of quinidine than that in the presence of vehicle both in normal and $Mg^{2+}$-free medium (Fig. 2c, d), while those were slightly smaller in the presence of vehicle (DMSO, final concentration 0.5%) than in the absence of vehicle (compare Fig. 2a and c). This result indicates that the $Na^+/Mg^{2+}$ exchanger does not mediate $H_2O_2$-induced $Mg^{2+}$ responses and that inhibition of $Mg^{2+}$ efflux retains $Mg^{2+}$ released from intracellular $Mg^{2+}$ sources.

Most of the intracellular $Mg^{2+}$ binds to various biomolecules, and a major binding partner of $Mg^{2+}$ in the cytoplasm is ATP. ATP normally binds to $Mg^{2+}$ in the form of an Mg–ATP complex, but the $Mg^{2+}$ is dissociated when ATP is consumed and degraded to ADP, leading to an increase in free $Mg^{2+}$. $Mg^{2+}$ dissociation from ATP due to ATP consumption has been reported as the cause of increases in $[Mg^{2+}]_{cyto}$ during mitosis and apoptosis[18,32]. To confirm the dissociation of $Mg^{2+}$ from ATP, the genetically encoded ATP sensor ATeam[33] was expressed in keratinocytes and the intracellular ATP level was visualized (Fig. 3a). It was confirmed that $H_2O_2$ does not directly affect ATeam signals independently of ATP (Supplementary Fig. 1b). Upon an application of $H_2O_2$, ATP concentration decreased in keratinocytes (Fig. 3b blue line). To examine the relationship between the decrease in ATP and increase in $[Mg^{2+}]_{cyto}$, $H_2O_2$-induced change in ATP concentration was also examined in the newborn keratinocytes. These cells showed

smaller increases in $H_2O_2$-induced $[Mg^{2+}]_{cyto}$ (Fig. 1e) and relatively smaller decreases in ATP compared to adult keratinocytes (Fig. 3b red line and c), whereas there were no significant difference in cellular ATP contents between those cells (Supplementary Fig. 4). We also examined whether the cells with large ATP decreases showed larger increases in $[Mg^{2+}]_{cyto}$. Mitochondria are a major source of cellular ATP, and their inhibition affects cellular ATP production. Therefore, the effects of prior inhibition of ATP production in the mitochondria on the $H_2O_2$-induced responses in ATP and $Mg^{2+}$ were investigated. Neither oligomycin, which is an inhibitor of $F_oF_1$ ATP synthase, nor FCCP alone elicited decreases in ATP levels at least within minutes of the application (Supplementary Fig. 5a, b). The efficacy of oligomycin and FCCP was confirmed by the results that when combined with the glycolysis inhibitor 2-deoxy-D-glucose (2DG), those induced greater decreases in cellular ATP levels than 2DG alone (Supplementary Fig. 5c, d). These results also indicate that mitochondria and glycolysis complement each other to maintain ATP concentration in keratinocytes. Interestingly, pretreatment with oligomycin or FCCP significantly enhanced $H_2O_2$-induced decreases in ATP (Fig. 3d, e). The inhibition of ATP synthesis in the mitochondria also enhanced the $H_2O_2$-induced increase in $[Mg^{2+}]_{cyto}$, although neither oligomycin alone nor FCCP alone induced increases in $[Mg^{2+}]_{cyto}$ (Fig. 3f, g), and $Mg^{2+}$ and ATP dynamics in response to $H_2O_2$ were mirror images of each other (compare Fig. 3d and f). These results indicate that $Mg^{2+}$ dissociation from ATP was the major $Mg^{2+}$ source in response to $H_2O_2$ in keratinocytes.

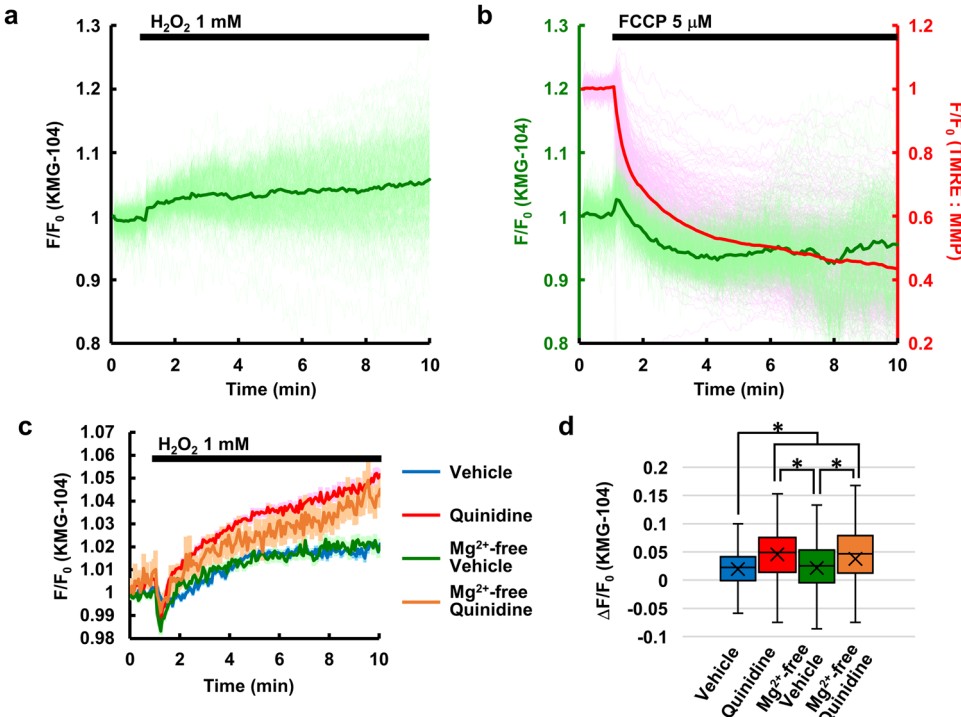

**Fig. 2 Examination of Mg²⁺ source in response to H₂O₂.** a $H_2O_2$-induced $Mg^{2+}$ response in $Mg^{2+}$-free HBSS. Mean (green line) and all traces (light green lines) ($n = 376$ cells from three different experiments). b Time-courses of $[Mg^{2+}]_{cyto}$ (green line indicates mean and light green lines indicate all traces, left axis) and mitochondrial membrane potential (MMP) (red line indicates mean and pink lines indicate all traces, right axis), measured simultaneously, in response to FCCP (5 μM) added at 1 min ($n = 394$ cells from three different experiments). c Average time-courses of $[Mg^{2+}]_{cyto}$ in response to $H_2O_2$ in the presence of vehicle (0.5% DMSO; blue line, $n = 512$ cells from four different experiments), quinidine (200 μM; red line, $n = 485$ cells from four different experiments), vehicle in $Mg^{2+}$-free condition (green line, $n = 340$ cells from three different experiments), and quinidine in $Mg^{2+}$-free condition (orange, $n = 389$ cells from three different experiments). Error bars: SEM. d Comparison of the average amplitude of $Mg^{2+}$ response shown in c. The amplitude was calculated as a difference between the average of $F/F_0$ before (0–1 min) and after (9–10 min) $H_2O_2$ treatment. Center line: median, x: average, box limits: quartiles, whiskers: 1.5× interquartile range. *$p < 0.05$ (Tukey's test, two-sided).

**$Mg^{2+}$ suppresses the $H_2O_2$-induced decrease in MMP in a concentration-dependent manner.** Our next question was whether increased $[Mg^{2+}]_{cyto}$ protected cells from oxidative stress and via what mechanism this occurred. It has been reported that $H_2O_2$ suppresses mitochondrial function by decreasing MMP[5,6]. On the other hand, previous studies show that $Mg^{2+}$ contributes to the maintenance of MMP and that increased $Mg^{2+}$ by $Mg^{2+}$ supplementation or inhibition of $Mg^{2+}$ efflux attenuates decreases in MMP under cellular stress conditions[20–22]. Therefore, the relationship between $[Mg^{2+}]_{cyto}$ and MMP was examined using simultaneous imaging with KMG-104 and TMRE (Fig. 4a). $H_2O_2$ caused an increase in $[Mg^{2+}]_{cyto}$ and a gradual decrease in MMP within 10 min (Fig. 4b). Keratinocytes that showed large $Mg^{2+}$ increases exhibited lower decreases in MMP (thin solid lines in Fig. 4b), and conversely, cells with small increases in $[Mg^{2+}]_{cyto}$ showed large decreases in MMP (thin dotted lines in Fig. 4b). To prove that $Mg^{2+}$ had a direct effect on $H_2O_2$-induced decreases in MMP, the cytosolic level of $Mg^{2+}$ was increased by adding $Mg^{2+}$ to the extracellular medium before $H_2O_2$ stimulation. Supplementation of $Mg^{2+}$ to the extracellular medium increased the $Mg^{2+}$ concentration in the medium from 0.9 to 5 mM and led to a steep increase in $[Mg^{2+}]_{cyto}$. The subsequent application of $H_2O_2$ did not induce significant changes in averaged $[Mg^{2+}]_{cyto}$, while some cells showed an increase or decrease in $[Mg^{2+}]_{cyto}$ (green line in Fig. 4c). Interestingly, $H_2O_2$-induced decreases in MMP were significantly prevented by the prior addition of $Mg^{2+}$ (Fig. 4d, e), whereas $Mg^{2+}$ supplementation itself had little effect on MMP (red line in Fig. 4c). These findings suggest that $[Mg^{2+}]_{cyto}$ affects MMP primarily under stress conditions. To estimate the relationship between $[Mg^{2+}]_{cyto}$ and $H_2O_2$-induced decreases in MMP, the changes in $[Mg^{2+}]_{cyto}$ from initial levels (shown as green lines in Fig. 4b, c) and the $H_2O_2$-induced changes in MMP that were normalized at 1 min before $H_2O_2$ application (from the data in Fig. 4d) in each cell were plotted (Fig. 4f). In both the normal condition (gray dots) and the high $Mg^{2+}$ condition (orange dots), strong correlations between $Mg^{2+}$ increases and MMP were observed; higher levels of cytosolic $Mg^{2+}$ were correlated with lower decreases in MMP. Interestingly, these two plots appear to line up on the same line, indicating that $[Mg^{2+}]_{cyto}$ is one of the key determinants for protecting the mitochondria from $H_2O_2$ damage (Fig. 4f).

Finally, we confirmed that the $Mg^{2+}$ supplementation suppresses the toxicity of $H_2O_2$. Exposure to 1 mM $H_2O_2$ for 24 h induced ~40% cell death in keratinocytes in a normal culture medium. Supplementation of additional 5 mM $Mg^{2+}$ to the culture medium suppressed the toxicity (Fig. 5). Our results indicate that $Mg^{2+}$ supplementation is effective in protecting keratinocytes from $H_2O_2$ toxicity.

## Discussion

The skin is continuously exposed to oxidative stress. This study revealed that $H_2O_2$, which is a kind of ROS, causes increases in $[Mg^{2+}]_{cyto}$ in human keratinocytes. This $H_2O_2$-induced $Mg^{2+}$ response was higher in adult keratinocytes than in newborn keratinocytes. Our findings indicate that the source of $Mg^{2+}$ was dissociation from ATP in the process of ATP consumption. Upon the addition of $H_2O_2$, decreases in MMP were also observed, and the change in MMP was strongly correlated with increases in

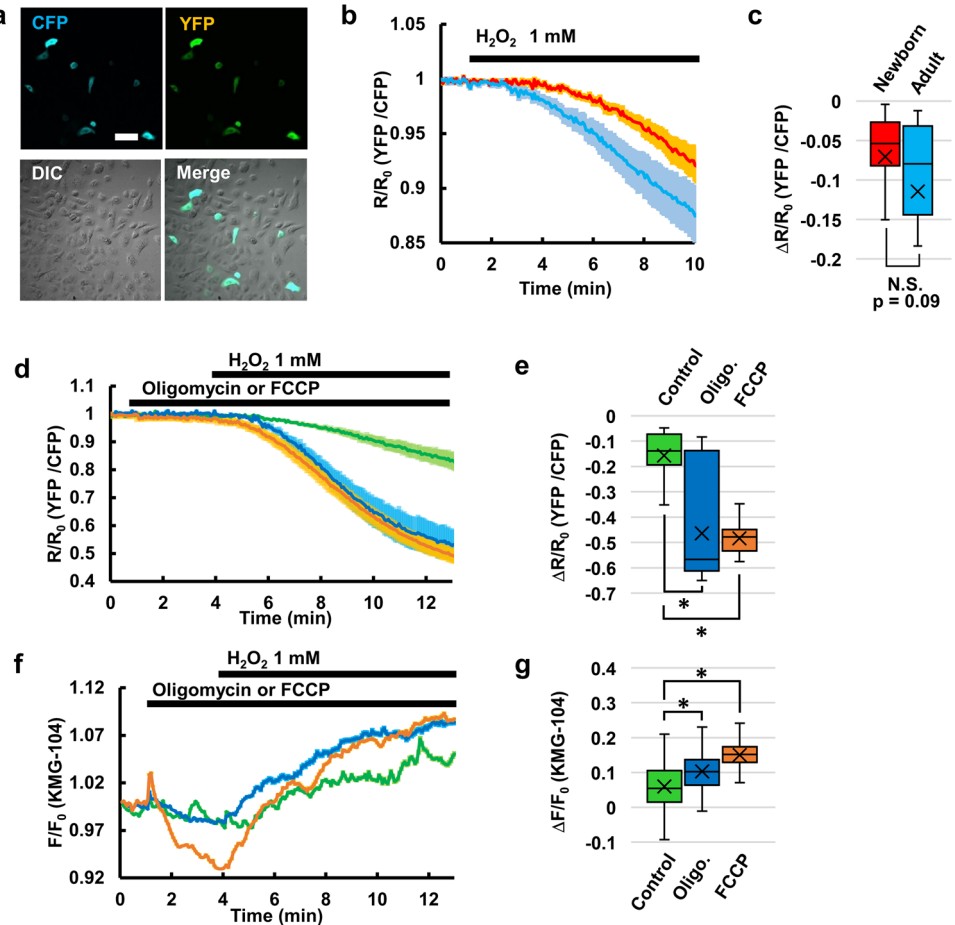

**Fig. 3 Inhibition of oxidative phosphorylation enhances $H_2O_2$-induced increases in $Mg^{2+}$ and decreases in ATP. a** Fluorescence images of keratinocytes expressing the ATP sensor ATeam (CFP, YFP, DIC, and Merge). Scale bar: 100 μm. **b** Average time-courses of ATP levels in newborn keratinocytes (red line: $n = 30$ cells from four different experiments) and adult keratinocytes (blue line: $n = 24$ cells from five different experiments) in response to $H_2O_2$ (1 mM) added at 1 min. Error bars: SEM. **c** Comparison of the average amplitude of $H_2O_2$-induced decreases in ATP in the newborn and adult keratinocytes shown in **b**. The amplitude was calculated as a difference between the average of $R/R_0$ before (0–1 min) and after (9–10 min) $H_2O_2$ treatment. N.S.: not significant (Student's $t$-test, two-sided). **d** Average time-courses of ATP levels in adult keratinocytes in response to the indicated inhibitors and subsequent $H_2O_2$. These treatments were as follows: control (green line: $n = 11$ cells from three different experiments), oligomycin (blue line: $n = 15$ cells from three different experiments), and FCCP (orange line: $n = 19$ cells from three different experiments). Depending on the treatment group, oligomycin (5 μM) or FCCP (5 μM) was added at 1 min and $H_2O_2$ was subsequently added at 4 min. Error bars: SEM. **e** The average amplitude of $H_2O_2$-induced decreases in ATP in the adult keratinocytes shown in **d**. The amplitude was calculated as a difference between the average of $R/R_0$ before (3–4 min) and after (12–13 min) $H_2O_2$ treatment. *$p < 0.05$ (Dunnett's test, two-sided). **f** Average time-courses of $Mg^{2+}$ response in adult keratinocytes in response to the indicated inhibitors and subsequent $H_2O_2$. These treatments were as follows: control (green line: $n = 426$ cells from three different experiments), Oligomycin (blue line: $n = 392$ cells from three different experiments), and FCCP (orange line: $n = 398$ cells from three different experiments). Depending on the treatment group, oligomycin (5 μM) or FCCP (5 μM) was added at 1 min and $H_2O_2$ was subsequently added at 4 min. Error bars: SEM. **g** The average amplitude of $H_2O_2$-induced $Mg^{2+}$ responses in the adult keratinocytes shown in **f**. The amplitude was calculated as a difference between the average of $F/F_0$ before (3–4 min) and after (12–13 min) $H_2O_2$ treatment. In the box plots in this figure, center line: median, $x$: average, box limits: quartiles, whiskers: 1.5× interquartile range. *$p < 0.05$ (Dunnett's test, two-sided).

$[Mg^{2+}]_{cyto}$. In other words, higher levels of cytosolic $Mg^{2+}$ were connected to lower MMP decreases. Furthermore, $H_2O_2$-induced decreases in MMP were significantly prevented by prior addition of $Mg^{2+}$, suggesting direct effects of $Mg^{2+}$ on the mitochondria. Moreover, $Mg^{2+}$ supplementation also suppressed $H_2O_2$-induced cell death. In summary, this study revealed that $H_2O_2$-induced $Mg^{2+}$ dissociation from ATP and that the resulting increase in $[Mg^{2+}]_{cyto}$ prevented MMP depolarization in keratinocytes. Our results suggest that $Mg^{2+}$ that has dissociated from ATP is not merely a byproduct, but functions as a cytoprotective mechanism against oxidative stress and that $Mg^{2+}$ supplementation is effective in protection against oxidative stress.

$Mg^{2+}$ mobilization from intracellular storage in keratinocytes has not yet been thoroughly investigated, while $Mg^{2+}$ influx via

NIPAL4, an $Mg^{2+}$ transporter, was reported previously[34–36]. In the present study, we demonstrated that $H_2O_2$ increases cytosolic free $Mg^{2+}$ by dissociation from ATP due to a decrease in cellular ATP level, although neither $Mg^{2+}$ influx, $Mg^{2+}$ transport via $Na^+/Mg^{2+}$ exchanger nor $Mg^{2+}$ release from the mitochondria was involved in this response. It has been reported that mitochondria are intracellular $Mg^{2+}$ storage sites and that depolarization of MMP induces $Mg^{2+}$ release from the mitochondria into the cytosol in other cell types[16,19,29,30]. In contrast, $[Mg^{2+}]_{cyto}$ instead decreased in response to FCCP in keratinocytes. This finding suggests that mitochondria do not act as $Mg^{2+}$ sources in keratinocytes, although they probably contain $Mg^{2+}$ in their matrix, similarly to other cell types, since some enzymes in the mitochondria require $Mg^{2+}$ for their activity[37,38]. ATP is

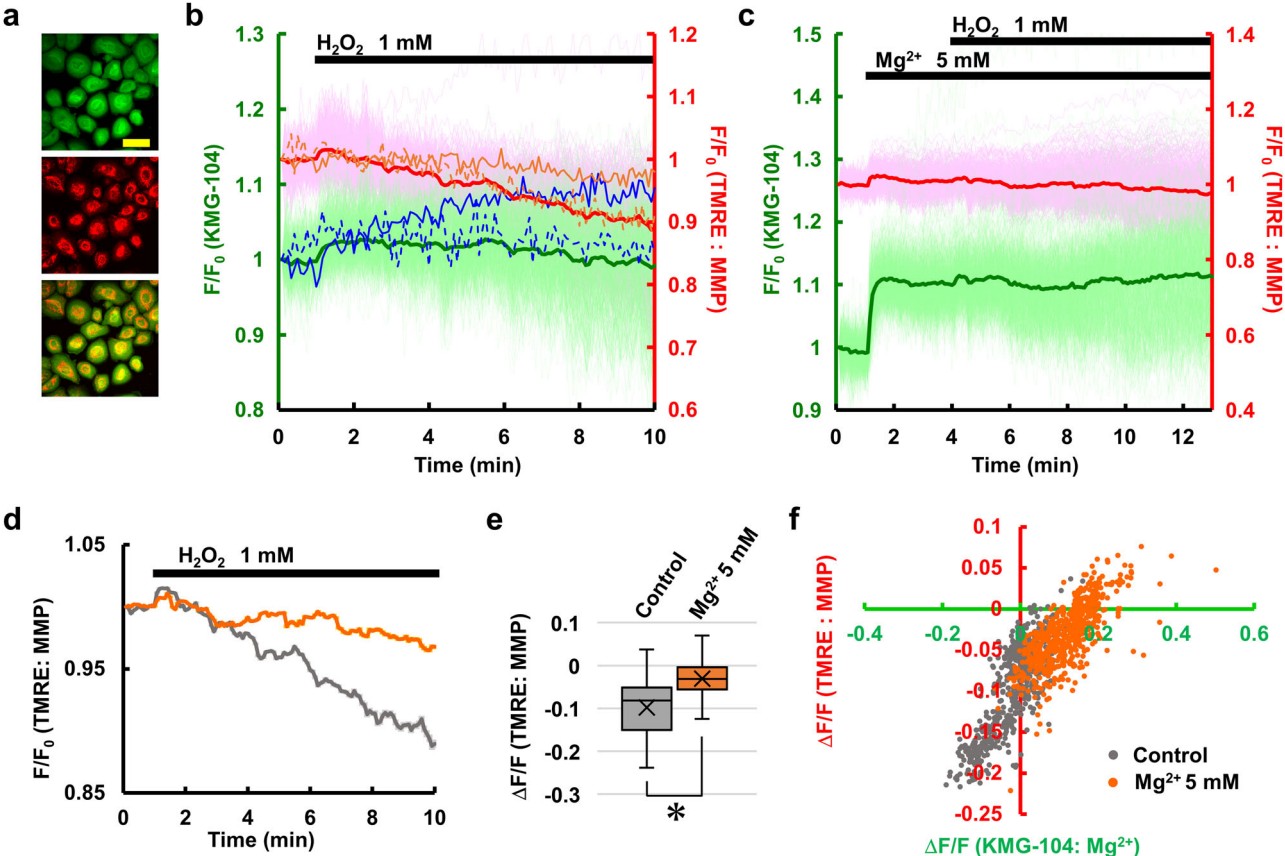

**Fig. 4 $H_2O_2$-induced decreases in MMP were suppressed by increases in $Mg^{2+}$. a** Fluorescence images of keratinocytes stained with KMG-104 (green) and TMRE (red), and a merged image. Scale bar: 100 μm. **b** Time-courses of $[Mg^{2+}]_{cyto}$ (green line indicates mean and light green lines indicate all traces, left axis) and MMP (red line indicates mean and pink lines indicate all traces, right axis), measured simultaneously, in response to $H_2O_2$ added at 1 min ($n = 542$ cells from three different experiments). Representative trace of normal responding cells is shown in blue ($[Mg^{2+}]_{cyto}$) and orange (MMP) dashed lines, and representative trace of cells with large $Mg^{2+}$ responses is shown in blue ($[Mg^{2+}]_{cyto}$) and orange (MMP) solid line. **c** Time-courses of $[Mg^{2+}]_{cyto}$ (green line indicates mean and light green lines indicate all traces, left axis) and MMP (red line indicates mean and pink lines indicate all traces, right axis) in response to stepwise increases in extracellular $Mg^{2+}$ concentration from 0.9 to 5 mM and the subsequent addition of $H_2O_2$ ($n = 581$ cells from three different experiments). **d** Average time-courses of $H_2O_2$-induced decreases in MMP under high $Mg^{2+}$ (5 mM) conditions (orange line: $n = 581$ cells from three different experiments) and that under normal conditions (control, gray line: $n = 542$ cells from three different experiments). Error bars: SEM. **e** Comparison of the amplitudes of $H_2O_2$-induced decreases in MMP that were shown in **d**. The amplitude was calculated as a difference between the average of $R/R_0$ before (0–1 min) and after (9–10 min) $H_2O_2$ treatment. Center line: median, x: average, box limits: quartiles, whiskers: 1.5× interquartile range. *$p < 0.05$. (Student's $t$-test, two-sided). **f** Scatter plot of the $H_2O_2$-induced increases in $[Mg^{2+}]_{cyto}$ ($\Delta F/F_0$) and decreases in MMP ($\Delta F/F_0$) in each cell that were shown in **d**. These responses were measured under high $Mg^{2+}$ conditions (orange plots) and normal conditions (gray plots).

known to be a major intracellular $Mg^{2+}$ binding partner: hundreds of enzymes utilize ATP in the form of Mg-ATP, and $Mg^{2+}$ is dissociated upon the degradation of ATP into ADP, increasing intracellular free $Mg^{2+}$[13,32,39]. Therefore, inhibition of mitochondrial ATP synthesis enhanced not only the decrease in ATP level but also the increase in $[Mg^{2+}]_{cyto}$ that was induced by $H_2O_2$. In contrast, newborn keratinocytes, which showed relatively smaller decreases in ATP compared to adult keratinocytes, had smaller $Mg^{2+}$ responses to $H_2O_2$ than adult keratinocytes. In the process of aging, the contribution of anaerobic respiration to the energy metabolism becomes substantial in keratinocytes[40]. This difference in the ATP production process between adult and newborn keratinocytes may cause larger ATP decreases and result in greater increases in $Mg^{2+}$ in response to $H_2O_2$ in adult keratinocytes compared to newborn keratinocytes. Our results show that keratinocytes that exhibited large decreases in ATP showed large $Mg^{2+}$ reactions in response to $H_2O_2$ and vice versa, strongly indicating that a major source of $Mg^{2+}$ under oxidative stress conditions is the dissociation of $Mg^{2+}$ from ATP.

In keratinocytes, oxidative stress changes metabolism drastically and acutely. Glucose usage changes from glycolysis to the pentose phosphate pathway within seconds of oxidative stress and decreased ATP levels occurrence[11]. Our study also focused on the acute response of keratinocytes to oxidative stress and demonstrated that reduced ATP levels lead to an increase in $Mg^{2+}$ levels. $Mg^{2+}$ has large effects on the cellular metabolism and also on mitochondrial functions. The positive effect of $Mg^{2+}$ on MMP under mitochondrial stress conditions has been demonstrated not only in isolated mitochondria but also in cells[20,21]. The contributions of $Mg^{2+}$ on MMP retention via inhibiting $K^+/H^+$ exchangers, preventing mitochondrial permeability transition pores (mPTP) from opening, and activating the tricarboxylic acid (TCA) cycle, has been reported previously[41]. In the present study, MMP levels under oxidative stress were also determined by $[Mg^{2+}]_{cyto}$, which was perturbed by oxidative stress. This result suggests that the change in $[Mg^{2+}]_{cyto}$ is a mitochondria-protective mechanism under oxidative stress conditions. Previous studies have also demonstrated the long-term effects of $Mg^{2+}$ on cell protection in keratinocytes[42] and other cells[22,24].

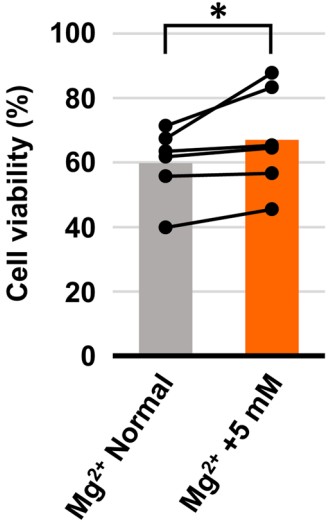

**Fig. 5 Mg$^{2+}$ supplementation attenuated H$_2$O$_2$-induced cell death.** Viability of the cells exposed to 1 mM H$_2$O$_2$ for 24 h in normal culture medium (gray) and culture medium supplemented with 5 mM Mg$^{2+}$ (orange). Bar graphs indicate average and each pair of dots connected line indicate data of each experiment ($n = 6$). *$p < 0.05$. (Student's $t$-test, one-sided).

$[Mg^{2+}]_{cyto}$ changes the activity of phosphatases, such as mTOR, CREB, and ERK[16], and leads to cell protection under stress conditions in keratinocytes[42] and neurons[22]. Therefore, Mg$^{2+}$ has both acute and long-term protective effects for cells.

Dissociation from ATP is a well-known source of Mg$^{2+}$. However, little has previously been known about the role of the dissociated Mg$^{2+}$. It has been reported that an increase in $[Mg^{2+}]_{cyto}$, which is probably dissociated from Mg-ATP, is required for DNA condensation during mitosis in HeLa cells[18], indicating that increased $[Mg^{2+}]_{cyto}$ resulting from ATP consumption is not a byproduct but instead plays an important role in cellular events. In the present study, we demonstrated that increasing levels of Mg$^{2+}$ during the process of ATP level decrease protected the mitochondria, an organelle that is responsible for ATP generation, under oxidative stress in keratinocytes. This suggest that Mg$^{2+}$ acts as a negative feedback signal to maintain ATP level against stress on cells. Some studies have already demonstrated that mitochondria was protected from oxidative stress under Mg$^{2+}$ rich conditions, but these studies have not referred to changes in $[Mg^{2+}]_{cyto}$ levels[24,27,42]. Our data reveal that $[Mg^{2+}]_{cyto}$ increases in response to oxidative stress and that Mg$^{2+}$ that dissociated from Mg-ATP is not a byproduct but instead acts as a mechanism to protect the mitochondria. Since the protective effect of Mg$^{2+}$ supplementation against oxidative stress has been reported in other cell types[23–27], the mechanism revealed here might be common in mammalian cells to protect cells against cellular stress.

In conclusion, we demonstrated that H$_2$O$_2$ induced an increase in $[Mg^{2+}]_{cyto}$ due to dissociation from Mg-ATP, and the increased $[Mg^{2+}]_{cyto}$ protected mitochondria from ROS damage. Moreover, supplementation of Mg$^{2+}$ to extracellular medium further suppressed the decrease in MMP and attenuated H$_2$O$_2$ toxicity. The skin is always exposed to oxidative stress from UV, so Mg$^{2+}$ would play an important role in maintaining the robustness of energy metabolism and protecting the skin from oxidative stress. Moreover, the addition of Mg$^{2+}$ from an external source showed an additive effect for cell protection, suggesting that Mg$^{2+}$ is a candidate active ingredient to protect skin from oxidative stress.

## Methods

**Cell cultures.** Normal human epidermal keratinocytes were purchased from Kurabo (Osaka, Japan). To compare the response of keratinocytes from newborn babies and adults, four different batches of newborn keratinocytes vials from different donors (0 years–1, 2, 3 and 4) and two different batches of adult keratinocytes vials (donors aged 40 years and 57 years) were used (details are summarized in Supplementary Table 1). Cells were cultured in EPILIFE$^{TM}$ medium (Thermo Fisher Scientific, Waltham, MA, USA) supplemented with insulin (10 µg mL$^{-1}$), human recombinant epidermal growth factor (0.1 ng mL$^{-1}$), hydrocortisone (0.67 µg mL$^{-1}$), gentamicin (50 µg mL$^{-1}$), amphotericin B (50 ng mL$^{-1}$), and bovine pituitary extract (0.4%, v/v), all of which were sourced from Kurabo, at 37 °C in a CO$_2$ incubator. Undifferentiated keratinocytes between passage 2 and passage 6 were used for experiments. EPILIFE$^{TM}$ medium contains Mg$^{2+}$, but the concentration is not disclosed.

For fluorescence imaging, keratinocytes were seeded on glass bottom dishes (IWAKI, Shizuoka, Japan) coated with 5 µg mL$^{-1}$ collagen (Sigma-Aldrich, Saint Louis, MO, USA) at a concentration of $6–8 \times 10^4$ cells mL$^{-1}$.

**Dye loading and fluorescence imaging.** For Mg$^{2+}$ imaging, cells were stained with an Mg$^{2+}$-selective fluorescent probe, KMG-104[43]. Keratinocytes were incubated with 20 µM KMG-104-AM and 200 µg mL$^{-1}$ Pluronic F-127 (Thermo Fisher Scientific) at 37 °C. After 30 min, the keratinocytes were washed twice with Ca$^{2+}$-free HBSS (Thermo Fisher Scientific; the pH was buffered using 10 mM HEPES and adjusted to 7.4 with NaOH, HBSS contains 0.9 mM Mg$^{2+}$) and incubated for a further 15 min in Ca$^{2+}$-free HBSS to allow the complete hydrolysis of acetoxy methyl (AM) groups. To avoid differentiation of keratinocytes, all experimental procedures were performed in Ca$^{2+}$-free medium.

For simultaneous imaging of cytosolic Mg$^{2+}$ and MMP, keratinocytes that had been loaded with KMG-104 were then incubated in Ca$^{2+}$-free HBSS containing 25 nM TMRE (Thermo Fisher Scientific) for 15 min at 37 °C. Fluorescence imaging was performed in Ca$^{2+}$-free HBSS with 2.5 nM TMRE.

A confocal laser scanning microscope system, FluoView FV1000 (Olympus, Tokyo, Japan), was used for the measurement of fluorescence. For the imaging of Mg$^{2+}$ alone, KMG-104 was excited at 488 nm using an Ar laser through a dichroic mirror (DM405/488, Olympus), and fluorescence at 510–610 nm was detected with a photomultiplier. Images were acquired every 4–6 s. For simultaneous imaging of cytosolic Mg$^{2+}$ and MMP, KMG-104 and TMRE were simultaneously excited at 488 nm using an Ar laser and 559 nm from a laser diode, respectively, through a dichroic mirror (DM405/488/559, Olympus). The emitted fluorescence was separated at 560 nm (SDM560, Olympus) and observed at 505–545 nm for KMG-104 and 570–670 nm for TMRE.

**Fluorescence imaging of intracellular ATP.** A fluorescence resonance energy transfer (FRET)-type ATP sensor, ATeam1.03[33], was kindly gifted from Dr. Imamura and was used to measure intracellular ATP levels. The plasmids that encoded ATeam were transfected into keratinocytes using Lipofectamine LTX (Thermo Fisher Scientific). These keratinocytes were cultured for 1–2 days after transfection to express the sensor proteins. Before observation, the cells were rinsed with and placed in Ca$^{2+}$-free HBSS.

The cells were observed on the confocal laser scanning microscope system Fluoview FV1000. ATeam was excited at 440 nm using a laser diode through a dichroic mirror (DM405–440/515, Olympus), and the emitted fluorescence was

separated by a dichroic mirror (SDM515, Olympus) and observed at 460–500 nm for CFP and at 515–615 nm for YFP.

For the negative control experiments, an ATP-insensitive variant of ATeam was constructed by inducing mutations of R122K and R126K in the ATP-sensing domain, following previous work[33]. The ATP-insensitive ATeam was expressed in the keratinocytes, and it was confirmed that $H_2O_2$ at a concentration below 10 mM had no impact on the fluorescent proteins.

**Image analysis**. The acquired images were analyzed using the software packages FluoView (Olympus), Aquacosmos (Hamamatsu Photonics, Shizuoka, Japan), and ImageJ. A region of interest (ROI) was assigned to the whole cell body of each cell, and the average fluorescence intensity in each ROI was calculated respectively. After subtracting the background, the time-course of fluorescence intensity for each cell ($F$) was normalized by the initial value ($F_0$), and the resulting $F/F_0$ values were compared between KMG-104 and TMRE. For ATeam, the ratio ($R$) of the fluorescence of cyan and yellow fluorescent protein (YFP/CFP) was calculated after subtracting background. The time-course of $R$ was normalized by the initial value ($R_0$), and the resulting $R/R_0$ values were compared.

Two groups of data were compared using Student's $t$-test. To compare multiple data sets, Dunnett's test or Tukey's tests were used. $P < 0.05$ was used to indicate significant differences.

**Measurement of $H_2O_2$ sensitivity of KMG-104 in vitro**. KMG-104 and several concentrations of $H_2O_2$ were mixed in 96-well plate and the fluorescence of these mixtures was measured using a Varioskan Flash spectral scanning multimode reader (Thermo Fisher Scientific). KMG-104 was excited at 500 nm and fluorescence intensity at 530 nm was measured and compared. $H_2O_2$ in the concentration range of 0–100 mM had no effect on the fluorescence of KMG-104 (Supplementary Fig. 1a).

**MTT assay**. Keratinocytes were plated at a density of $8.0 \times 10^3$ cells par well in a 96-well plate and incubated at 37 °C for more than 24 h. Medium was replaced to $Mg^{2+}$ normal medium (normal EPILIFE with supplements) or $Mg^{2+}$ +5 mM medium (EPILIFE with additional 5 mM $Mg^{2+}$ and supplements) 10 min prior to $H_2O_2$ application. The medium was replaced to $Mg^{2+}$ normal medium or $Mg^{2+}$ +5 mM medium containing 0 or 1 mM $H_2O_2$, and cells were incubated for 24 h in the incubator. Then, the medium was replaced to 0.5 mg mL$^{-1}$ MTT containing culture medium, and the cells were incubated for 2 h. The medium was discarded, and DMSO was then added to each well to dissolve the precipitate. The absorption at 575 nm was measured on a microplate reader, Valioscan (Thermo Fisher Scientific). The viabilities were calculated as a ratio to the average of $H_2O_2$ 0 mM condition for each $Mg^{2+}$ concentration.

**Statistics and reproducibility**. Fluorescence imaging experiments were repeated for 3–4 times for each experiment, and response of all the cells emitting sensor fluorescence in the field of view were analyzed. MTT assay was repeated for six times. Student's $t$-test was used for comparison of a pair of data. Dannett's test was used to compare multiple groups to control group. Tukey's test was used to compare all differences among data group more than three groups.

**Reporting summary**. Further information on research design is available in the Nature Portfolio Reporting Summary linked to this article.

**Data availability**

All data used in this study have been uploaded as a supplementary file named "Supplementary data 1".

**Code availability**

FluoView (Olympus), Aquacosmos (Hamamatsu Photonics, Shizuoka, Japan), and ImageJ were used for process and analyze image data.

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

## Acknowledgements
The plasmid encoding ATeam was kindly gifted from Dr. Imamura (Kyoto University).

## Author contributions
Conceptualization, K.O. and Y.K.; Investigation, K.F., Y.S., and M.G.; Writing—original draft, K.F. and Y.S.; Writing—review & editing, Y.K., M.G., K.H., and K.O.; Supervision, Y.K. and K.O.

## Competing interests
The authors declare no competing interests.
