## [Peer Review File · Communications Biology]

Reviewers' comments:

Reviewer #1 (Remarks to the Author):

In the present study, the authors investigated the spatio-temporal dynamics of intracellular Mg²⁺ in protecting mitochondria in keratinocytes in response to H₂O₂ administration. IN the experimental model tested in the study, inhibition of mitochondrial ATP synthesis enhanced teh H₂O₂ induced Mg²⁺ response, indication that most of Mg²⁺ mobilized by H₂O₂ was derived from the dissociation from ATP. Artificially increasing cytosolic Mg²⁺ resulted in an attenuated decrease in MMP decline in response to H₂O₂. The conclusion of the authors is that Mg²⁺ dissociated from ATP acts as a protective mechanism in mitochondria exposed to oxidative stress.

Comments

The study addresses the importance of Mg²⁺ in modulating and possibly protecting keratinocytes under oxidative stress.

A couple of points need further clarification.

1. The authors measured changes in cytosolic Mg²⁺ under various stimulatory conditions including increasing [Mg²⁺]_o from less than 1 mM to 5 mM. What was the change in cellular ATP under those experimental conditions (fig. 4)? Is it possible that increasing cytosolic Mg²⁺ can play a protective role on cellular ATP content? Also, it is possible that some magnesium may leave the cells under ~1 mM [Mg²⁺]_o but not under ~5 mM [Mg²⁺]_o?

2. Why did the authors focus on MMP (Fig 4) when the results in Fig. 2 indicated that mitochondria were NOT the source of the decrease in ATP?

3. Also, it is my understanding that changes in mitochondrial membrane potential in adult keratinocytes are an indication of keratinocytes aging process. Hence it is unclear to this reviewer whether this is the underlying reason the authors selected adult keratinocytes for the main experiments reported in the study as compared to newborn keratinocytes (Fig. 1), which showed limited or no response to H₂O₂ exposure in terms of Mg²⁺ changes AND ATP changes (Fig. 3). In this regard, was the total ATP content similar between newborn and adult cell lines? The author reports the results as changes in ratio but both cell lines start at 1 in Fig. 3B, which not necessarily means that the total ATP content is the same in absolute terms, only that the ratiometric decrease is smaller in newborn, but this could perhaps be the result of a smaller cell volume in newborn cells, and consequently a reduced cell loading with the various indicators.

Reviewer #2 (Remarks to the Author):

Present manuscript describes protective effect of Mg²⁺ on H₂O₂-induced decrease of mitochondrial membrane potential. It seems to be interesting and well performed. Anyway, some issues have to be clarified before acceptance.

Major comment:

H₂O₂-induced Increase of Mg²⁺ concentration was attributed to the decrease of ATP concentration. FCCP as uncoupler can also induce decrease of ATP level. There is a putative discrepancy associated with fact that FCCP alone induces decrease of Mg while FCCP in combination with H₂O₂ induces increase of Mg. Authors claim that neither oligomycin as an inhibitor of Fo-F1 ATP synthase nor FCCP alone elicited decreases in ATP level. From my personal experience, I know that uncouplers are decreasing intracellular ATP concentration but the process is not fast enough to lead to the significant ATP decrease within 5 minutes. Anyway, authors should show impact of FCCP alone on ATP concentration. In addition, there is just correlation between decreased ATP level and increased cytosolic Mg²⁺ concentration. Although it sounds reliable, another mechanism of Mg concentration

changes could be involved (e.g. impact of H₂O₂ on intracellular magnesium transporters).

Minor comment:

It is not common to refer to figures within discussion.

Reviewer #3 (Remarks to the Author):

Fujita et al. revealed the new function of Mg²⁺ dissociated from ATP in the protection of mitochondria under oxidative stress. The phenomena are interesting, but the data are preliminary.

Comments,

1. lines 76 and 86, Do EPILIFE medium and HBSS contain Mg²⁺?
2. line 89, Why did you use undifferentiated keratinocytes?
3. line 145, The amplitudes of the H₂O-induced Mg²⁺ response were different in between newborn and adult keratinocytes. You should describe the reason. In the first place, why did you compare the response between newborn and adult keratinocytes?
4. line 157, Are there any possibility that H₂O₂ blocked Mg²⁺ efflux from the cells?
5. line 175, You described neither oligomycin nor FCCP alone elicited decrease in ATP level or increase in [Mg²⁺]_{cyto}. Was the function of mitochondria blocked by these chemicals?
6. line 175, Why were the H₂O₂-elicited responses enhanced by oligomycin or FCCP? Was the binding force between ATP and Mg²⁺ changed by these chemicals?
7. Did you check the protective effect of Mg²⁺ against H₂O₂-induced toxicity under your experimental conditions?

Answers to Reviewers

Reviewer #1 (Remarks to the Author):

In the present study, the authors investigated the spatio-temporal dynamics of intracellular Mg^{2+} in protecting mitochondria in keratinocytes in response to H_2O_2 administration. In the experimental model tested in the study, inhibition of mitochondrial ATP synthesis enhanced the H_2O_2 induced Mg^{2+} response, indication that most of Mg^{2+} mobilized by H_2O_2 was derived from the dissociation from ATP. Artificially increasing cytosolic Mg^{2+} resulted in an attenuated decrease in MMP decline in response to H_2O_2 . The conclusion of the authors is that Mg^{2+} dissociated from ATP acts as a protective mechanism in mitochondria exposed to oxidative stress.

Comments

The study addresses the importance of Mg^{2+} in modulating and possibly protecting keratinocytes under oxidative stress. A couple of points need further clarification.

1. The authors measured changes in cytosolic Mg^{2+} under various stimulatory conditions including increasing $[Mg^{2+}]_o$ from less than 1 mM to 5 mM. What was the change in cellular ATP under those experimental conditions (fig. 4)? Is it possible that increasing cytosolic Mg^{2+} can play a protective role on cellular ATP content? Also, it is possible that some magnesium may leave the cells under ~ 1 mM $[Mg^{2+}]_o$ but not under ~ 5 mM $[Mg^{2+}]_o$?

We found the slight decrease in cellular ATP concentration upon an increase in $[Mg^{2+}]_o$ from 0.9 to 1 mM (Figure for reviewer 1). Mg^{2+} could affect not only a protection of mitochondria from stresses and an upregulation of cellular ATP synthesis but also an activation of cellular ATP consumption. Our result suggests that increased $[Mg^{2+}]_{cyto}$ upregulates ATP consumption, rather than ATP synthesis, in the keratinocytes under normal conditions. Actually, previous studies show that Mg^{2+} activates enzymes, some intracellular signals and cellular metabolism (reviewed by Yamanaka et al., *Int. J. Mol. Sci.*, 2019; Yamanaka et al., *Curr. Biol.*, 2018; Li et al., *Nature*, 2011; Feeney et al., *Nature*, 2016).

Figure for reviewer 1. An impact of increase in extracellular Mg^{2+} concentration from 0.9 to 5 mM on cellular ATP level.

(A) Average time-course of change in cellular ATP level in adult keratinocyte measured with ATeam in response to an increase in extracellular Mg^{2+} concentration from 0.9 to 5 mM ($n = 52$ cells from 3 different experiments). (B) Comparison

of the average ratio of ATeam (R/R_0) before (0–1 min) and after (9–10 min) increase in extracellular Mg^{2+} concentration. *: $p < 0.05$ (Student's t-test, Two-sided). Error bars in this figure: S.E.M.

Next, we checked whether Mg^{2+} leave the cells after the H_2O_2 -induced increase in $[Mg^{2+}]_{cyto}$. In many cell types, Mg^{2+} efflux via Na^+/Mg^{2+} exchanger, which is inhibited by quinidine, has been reported (Reference No. 14, 22, 31 in our revised manuscript). So, the effect of quinidine on the H_2O_2 -induced change in $[Mg^{2+}]_{cyto}$ was examined. If Mg^{2+} leaves the cells via the Na^+/Mg^{2+} exchanger, the amplitude of H_2O_2 -induced Mg^{2+} response is expected to be enhanced in the presence of quinidine. As expected, the responses were enhanced in the presence of quinidine under both normal and Mg^{2+} -free conditions (Figures added to Fig. 2). This indicates that some of the Mg^{2+} released from Mg-ATP in response to H_2O_2 left the cells via the Na^+/Mg^{2+} exchanger.

Figures added to Fig. 2.

(C) Average time-course of $[Mg^{2+}]_{cyto}$ in response to H_2O_2 in the presence of vehicle (0.5% DMSO; blue line, $n = 512$ cells from 4 different experiments), quinidine (200 μM ; red line, $n = 485$ cells from 4 different experiments), vehicle in Mg^{2+} -free condition (green line, $n = 340$ cells from 3 different experiments), and quinidine in Mg^{2+} -free condition (orange, $n = 389$ cells from 3 different experiments). (D) Comparison of the average amplitude of Mg^{2+} response shown in (C). The amplitude was calculated as a difference between the average of F/F_0 before (0–1 min) and after (9–10 min) H_2O_2 treatment. *: $p < 0.05$. (Tukey's test, two-sided). Error bars in this figure: S.E.M.

Then, we also compared the time-courses of H_2O_2 -induced change in $[Mg^{2+}]_{cyto}$ under $[Mg^{2+}]_o$ at 0.9 mM and 5 mM. The data of Mg^{2+} in Figure 4 was normalized to the values 1 min before H_2O_2 application and compared. As you mentioned, the shapes of graphs look slightly different as shown in Figure for reviewer 2. This suggest that the Mg^{2+} release was affected by the difference in $[Mg^{2+}]_o$.

Figure for reviewer 2. Comparison of H₂O₂-induced Mg²⁺ response under Mg²⁺ 0.9 mM and Mg²⁺ 5 mM.

2. Why did the authors focus on MMP (Fig 4) when the results in Fig. 2 indicated that mitochondria were NOT the source of the decrease in ATP?

Fig. 2 shows mitochondria were not the source of Mg²⁺ in response to H₂O₂. And, “that mitochondria were NOT the source of the decrease in ATP” was shown in Fig. 3. So, we could not decide whether your question meant “~ the source of **the decrease in ATP?**” or “~ the source of **the increase in Mg²⁺?**”.

Anyway, in Fig. 4, we attempted to investigate whether the increased Mg²⁺ plays any role. Some studies have demonstrated that H₂O₂ attacks mitochondria and induces decrease in their membrane potential (reference No. 5 and 6 in our manuscript). Actually, H₂O₂ reduced mitochondrial membrane potential (MMP) as shown Fig. 4. As you mentioned, this reduction in MMP had no effect on cellular ATP and Mg²⁺ levels at least within minutes. On the other hand, our group and also other groups have demonstrated a protective effect of Mg²⁺ on MMP in various types of cells (reference No. 12, 23–27 in our manuscript). Therefore, we hypothesized that the increased Mg²⁺ due to a release from Mg-ATP may have some positive effect on MMP. For this reason, we decided to check the mitochondrial membrane potential in Fig. 4.

3. Also, it is my understanding that changes in mitochondrial membrane potential in adult keratinocytes are an indication of keratinocytes aging process. Hence it is unclear to this reviewer whether this is the underlying reason the authors selected adult keratinocytes for the main experiments reported in the study as compared to newborn keratinocytes (Fig. 1), which showed limited or no response to H₂O₂ exposure in terms of Mg²⁺ changes AND ATP changes (Fig. 3). In this regard, was the total ATP content similar between newborn and adult cell lines? The author reports the results as changes in ratio but both cell lines start at 1 in Fig. 3B, which not necessarily means that the total ATP content is the same in absolute terms, only that the ratiometric decrease is smaller in newborn, but this could perhaps be the result of a smaller cell volume in newborn cells, and consequently a reduced cell loading with the various indicators.

At first, we found that H₂O₂ induced an increase in [Mg²⁺]_{cyto} in adult keratinocytes. To confirm that H₂O₂-induced increase in [Mg²⁺]_{cyto} is a common phenomenon across individual differences or age, we compared the response in keratinocytes from newborns and adults. Interestingly, the average responses varied by cell line, and adult keratinocytes exhibited greater Mg²⁺ increase in response to H₂O₂ than newborn keratinocytes (Fig. 1 and Supplementary Figure 1). Because we are interested in the underlying mechanism of the change in [Mg²⁺]_{cyto} and the roles of the Mg²⁺, in this study, we focused on the cells with large Mg²⁺ response to H₂O₂, in this case adult keratinocytes. As you mentioned, aging relates to mitochondria. In addition, we have demonstrated that mitochondria are profoundly relate to changes in cellular Mg²⁺ concentration in neurons (Shindo *et al.*, *J Neurosci Res*, 2010; Yamanaka *et al.*, *FEBS Lett*, 2013; Yamanaka *et al.*, *Curr Biol*, 2018). However, the Mg²⁺ source in response to H₂O₂ in keratinocytes was not the mitochondria.

We measured total ATP content in keratinocyte and compared by luciferin-luciferase assay (We had run out of 0 years-3 cell line, so we used 0 years-4 cell line instead). Cellular total ATP content varied with cell line, but there seemed to be any significant tendency with donor age (Supplementary Figure 4). These results indicate that the difference in H₂O₂-induced ATP decreases between newborn and adult keratinocytes was not due to the difference in their ATP contents. These results have been added as Supplementary Figure 4.

Supplementary Figure 4. ATP content in each keratinocyte line.

(A) Cellular APT contents in three newborn keratinocyte cell lines and two adult keratinocyte cell line measured by luciferin-luciferase assay (n = 5 for each). (B) Comparison of cellular ATP contents between newborn group and adult group. The data in (A) were divided into these two groups and compared. Error bars: S.E.M. N.S.: there was no significant difference (left: Tukey's test, Two-sided; right: Student's t-test, Two-sided).

Reviewer #2 (Remarks to the Author):

Present manuscript describes protective effect of Mg^{2+} on H_2O_2 -induced decrease of mitochondrial membrane potential. It seems to be interesting and well performed. Anyway, some issues have to be clarified before acceptance.

Major comment:

H_2O_2 -induced Increase of Mg^{2+} concentration was attributed to the decrease of ATP concentration. FCCP as uncoupler can also induce decrease of ATP level. There is a putative discrepancy associated with fact that FCCP alone induces decrease of Mg while FCCP in combination with H_2O_2 induces increase of Mg. Authors claim that neither oligomycin as an inhibitor of Fo-F1 ATP synthase nor FCCP alone elicited decreases in ATP level. From my personal experience, I know that uncouplers are decreasing intracellular ATP concentration but the process is not fast enough to lead to the significant ATP decrease within 5 minutes. Anyway, authors should show impact of FCCP alone on ATP concentration.

We examined changes in ATP concentration in response to FCCP alone or oligomycin alone using ATeam. As shown below, FCCP induced gradual decrease in cellular ATP concentration, but the response was slow and small. The decrease was not significant within 5 min, while a statistically significant (but small) decrease was observed within 10 min. On the other hand, oligomycin alone did not induced any significant change in cellular ATP concentration within 10 min. These results have been added as Supplementary Figure 5.

Supplementary Figure 5. Effect of oligomycin and FCCP on cellular ATP level

(A) Average time-courses ATP levels in adult keratinocytes measured with ATeam in response to FCCP (5 μ M, orange line: n = 43 cells from 3 different experiments) or oligomycin (5 μ M, light blue line: n = 41 cells from 3 different experiments). (B) Comparison of average ratio of the data in (A) at 0–1 min, 4–5 min, and 9–10 min. *: $p < 0.05$ (Dunnett's test, Two-sided). Error bars in this figure: S.E.M.

In addition, there is just correlation between decreased ATP level and increased cytosolic Mg^{2+} concentration. Although it sounds reliable, another mechanism of Mg concentration changes could be involved (e.g. impact of H_2O_2 on intracellular magnesium transporters).

We investigated the involvement of Mg^{2+} transporters in the changes in $[Mg^{2+}]_{cyto}$ in response to H_2O_2 . We have already demonstrated that Mg^{2+} uptake from extracellular medium was not required for H_2O_2 -induced increase in $[Mg^{2+}]_{cyto}$ in

Figure 2A. So, we checked an involvement of Mg^{2+} release from keratinocytes. It has been reported that Na^+/Mg^{2+} exchanger, one of which is SLC41A1, mediates Mg^{2+} efflux from the cells and are inhibited by quinidine (Reference No. 14, 22, 31 in our revised manuscript). Therefore, if one of the targets of H_2O_2 was the Na^+/Mg^{2+} exchanger, which led to the increase in $[Mg^{2+}]_{cyto}$, it is expected that quinidine also increased an increase in $[Mg^{2+}]_{cyto}$ and that prior application of quinidine abolished H_2O_2 -induced increase in $[Mg^{2+}]_{cyto}$. Quinidine alone induced increase in $[Mg^{2+}]_{cyto}$ in both normal and Mg^{2+} -free conditions, whereas vehicle (DMSO, final concentration 0.5%) decreased $[Mg^{2+}]_{cyto}$, indicating that inhibition of Mg^{2+} efflux leads to increase in $[Mg^{2+}]_{cyto}$ in keratinocytes (newly added Supplementary Figure 3A). The amplitude of quinidine-induced Mg^{2+} increase was greater in normal condition than Mg^{2+} -free condition (newly added Supplementary Figure 3B). These results suggest that $[Mg^{2+}]_{cyto}$ is normally balanced by Mg^{2+} influx and efflux. The effect of quinidine on H_2O_2 -induced Mg^{2+} responses was also investigated. The responses were greater in the presence of quinidine than that in the presence of vehicle both in normal and Mg^{2+} -free medium (newly added Fig. 2C and D), while those were slightly smaller in the presence of vehicle (DMSO, final concentration 0.5%) than in the absence of vehicle (compare Fig. 2A and C). These results indicate that the Na^+/Mg^{2+} exchanger does not mediate H_2O_2 -induced Mg^{2+} responses and that inhibition Mg^{2+} efflux retains Mg^{2+} released from intracellular Mg^{2+} sources in cytoplasm. The explanation above has been added to main text and the Figures shown below have been added to Fig. 2 and Supplementary Figure 3.

Supplementary Figure 3. Quinidine induced increases in $[Mg^{2+}]_{cyto}$

(A) Average time course of $[Mg^{2+}]_{cyto}$ in response to vehicle (DMSO, final concentration 0.5%; blue line, $n = 512$ cells from 4 different experiments), quinidine (200 μ M; red line, $n = 488$ cells from 4 different experiments), vehicle in Mg^{2+} -free condition (green line, $n = 340$ cells from 3 different experiments), and quinidine in Mg^{2+} -free condition (orange, $n = 396$ cells from 3 different experiments). Quinidine or vehicle was added at 1 min by bath application. (B) Comparison of the average amplitude of Mg^{2+} response shown in (A). The amplitude was calculated as a difference between the average of F/F_0 before (0–1 min) and after (9–10 min) stimulus. *: $p < 0.05$. (Tukey's test, two-sided). Error bars in this figure: S.E.M.

Figures added to Fig. 2

(C) Average time-course of $[Mg^{2+}]_{cyto}$ in response to H_2O_2 in the presence of vehicle (0.5% DMSO; blue line, $n = 512$ cells from 4 different experiments), quinidine (200 μM ; red line, $n = 485$ cells from 4 different experiments), vehicle in Mg^{2+} -free condition (green line, $n = 340$ cells from 3 different experiments), and quinidine in Mg^{2+} -free condition (orange, $n = 389$ cells from 3 different experiments). (D) Comparison of the average amplitude of Mg^{2+} response shown in (C). The amplitude was calculated as a difference between the average of F/F_0 before (0–1 min) and after (9–10 min) H_2O_2 treatment. *: $p < 0.05$. (Tukey's test, two-sided). Error bars in this figure: S.E.M.

Minor comment:

It is not common to refer to figures within discussion.

Thank you for your advice. We removed figure references from the discussion section.

Reviewer #3 (Remarks to the Author):

Fujita et al. revealed the new function of Mg²⁺ dissociated from ATP in the protection of mitochondria under oxidative stress. The phenomena are interesting, but the data are preliminary.

Comments,

1. lines 76 and 86, Do EPILIFE medium and HBSS contain Mg²⁺?

Both EPILIFE and HBSS contains Mg²⁺. But, Mg²⁺ concentration in EPILIFE is not disclosed. HBSS contains 0.9 mM Mg²⁺. This information was added to the Methods section.

2. line 89, Why did you use undifferentiated keratinocytes?

There are many studies using undifferentiated keratinocytes as well as studies using differentiated keratinocytes. So our sense is that it is also normal to use undifferentiated keratinocytes. Of course, we confirmed that differentiation did not alter H₂O₂-induced Mg²⁺ responses (data not shown).

3. line 145, The amplitudes of the H₂O₂-induced Mg²⁺ response were different in between newborn and adult keratinocytes. You should describe the reason. In the first place, why did you compare the response between newborn and adult keratinocytes?

At first, we found that H₂O₂ induces increase in Mg²⁺ concentration in adult keratinocytes (keratinocytes from 40 years old). To confirm that H₂O₂-induced increase in [Mg²⁺]_{cyto} is a common phenomenon across individual differences or age, we compared the responses in keratinocytes from newborns and adults. This is the reason why we used both newborn and adult keratinocytes. Interestingly, the average responses varied by cell line, and adult keratinocytes exhibited greater Mg²⁺ responses to H₂O₂ than newborn keratinocytes as shown Fig. 1 and Supplementary Figure 1.

We have added some phrases and sentences to the Results section to explain the above.

In this study, we demonstrated that H₂O₂ induces decrease in ATP concentration, which leads to release of Mg²⁺ from Mg-ATP complex and increase in [Mg²⁺]_{cyto}. (In the cytosol, ATP normally binds Mg²⁺ to forms Mg-ATP complex. Since proteins usually consume ATP in the form of Mg-ATP, ATP consumption leases to an increase in free Mg²⁺ dissociated from the Mg-ATP complex.) We concluded that this is the mechanism that H₂O₂ induce the increase in [Mg²⁺]_{cyto}. Therefore, the answer to the question “Why the amplitudes of the H₂O₂-induced Mg²⁺ responses were different between newborn and adult keratinocytes?” is that H₂O₂ tends to cause a greater decrease in ATP concentration in adult keratinocytes than in newborn keratinocytes. We demonstrated this in the Figure 2 and 3 in our manuscript. It is not elucidated in this study why H₂O₂ causes a greater decrease in ATP concentration in adult keratinocytes than in newborn keratinocytes, but it appears to be outside the scope of this study.

4. line 157, Are there any possibility that H₂O₂ blocked Mg²⁺ efflux from the cells?

Thank you for your sharp question. We checked this possibility. It has been reported that Na⁺/Mg²⁺ exchanger, one of which is SLC41A1, mediates Mg²⁺ efflux from the cells and are inhibited by quinidine (Reference No. 14, 22, 31 in our revised manuscript). Therefore, if one of the targets of H₂O₂ is the Na⁺/Mg²⁺ exchanger, it is expected that quinidine also increased an increase in [Mg²⁺]_{cyto} and that prior application of quinidine abolished H₂O₂-induced increase in [Mg²⁺]_{cyto}. Quinidine alone induced increase in [Mg²⁺]_{cyto} in both normal and Mg²⁺-free conditions, whereas vehicle (DMSO, final concentration 0.5%) decreased [Mg²⁺]_{cyto}, indicating that inhibition of Mg²⁺ efflux leads to increase in [Mg²⁺]_{cyto} in keratinocytes (newly added Supplementary Figure 3A). The amplitude of quinidine-induced Mg²⁺ increase was greater in normal condition than Mg²⁺-free condition (newly added Supplementary Figure 3B). These results suggest that [Mg²⁺]_{cyto} is normally balanced by Mg²⁺ influx and efflux. The effect of quinidine on H₂O₂-induced Mg²⁺ responses was also investigated. The responses were greater in the presence of quinidine than that in the presence of vehicle both in normal and Mg²⁺-free medium (newly added Fig. 2C and D), while those were slightly smaller in the presence of vehicle (DMSO, final concentration 0.5%) than in the absence of vehicle (compare Fig. 2A and C). These results indicate that the Na⁺/Mg²⁺ exchanger does not mediate H₂O₂-induced Mg²⁺ responses and that inhibition of Mg²⁺ efflux retains Mg²⁺ released from intracellular Mg²⁺ sources in cytoplasm.

The explanation above has been added to main text and the Figures shown below has been added to Fig. 2 and Supplementary Figure 3.

Supplementary Figure 3. Quinidine induced increases in [Mg²⁺]_{cyto}

(A) Average time course of [Mg²⁺]_{cyto} in response to vehicle (DMSO, final concentration 0.5%; blue line, n = 512 cells from 4 different experiments), quinidine (200 μM; red line, n = 488 cells from 4 different experiments), vehicle in Mg²⁺-free condition (green line, n = 340 cells from 3 different experiments), and quinidine in Mg²⁺-free condition (orange, n = 396 cells from 3 different experiments). Quinidine or vehicle was added at 1 min by bath application. (B) Comparison of the average amplitude of Mg²⁺ response shown in (A). The amplitude was calculated as a difference between the average of F/F₀ before (0–1 min) and after (9–10 min) stimulus. *: *p* < 0.05. (Tukey's test, two-sided). Error bars in this figure: S.E.M.

Figures added to Fig. 2

(C) Average time-course of $[Mg^{2+}]_{cyto}$ in response to H_2O_2 in the presence of vehicle (0.5% DMSO; blue line, $n = 512$ cells from 4 different experiments), quinidine (200 μM ; red line, $n = 485$ cells from 4 different experiments), vehicle in Mg^{2+} -free condition (green line, $n = 340$ cells from 3 different experiments), and quinidine in Mg^{2+} -free condition (orange, $n = 389$ cells from 3 different experiments). (D) Comparison of the average amplitude of Mg^{2+} response shown in (C). The amplitude was calculated as a difference between the average of F/F_0 before (0–1 min) and after (9–10 min) H_2O_2 treatment. *: $p < 0.05$. (Tukey's test, two-sided). Error bars in this figure: S.E.M.

5. line 175, You described neither oligomycin nor FCCP alone elicited decrease in ATP level or increase in $[Mg^{2+}]_{cyto}$. Was the function of mitochondria blocked by these chemicals?

In keratinocytes, FCCP or oligomycin alone did not decrease cellular ATP concentration immediately after application. As shown in figures below, while FCCP slightly decreased ATP within 10 min, oligomycin had no effect on cellular ATP concentration. It is probably due to the complementation by glycolysis. Inhibition of glycolysis by 2DG alone also did not affect cellular ATP concentration. However, combination use of 2DG with FCCP or oligomycin elicited significant decrease in ATP concentration. These results indicate that FCCP and oligomycin certainly inhibited mitochondrial ATP synthesis and that mitochondria and glycolysis compensate each other to maintain ATP level in keratinocytes.

The following figures were added to our manuscript as Supplementary Figure 5.

Supplementary Figure 5. Effect of oligomycin and FCCP on cellular ATP level

(A) Average time-courses ATP levels in adult keratinocytes measured with ATeam in response to FCCP (5 μ M, orange line: $n = 43$ cells from 3 different experiments) or oligomycin (5 μ M, light blue line: $n = 41$ cells from 3 different experiments). (B) Comparison of average ratio of the data in (A) at 0–1 min, 4–5 min, and 9–10 min. *: $p < 0.05$ (Dunnnett's test, Two-sided). (C) Average time-courses ATP levels in adult keratinocytes measured with ATeam in response to 2DG (10 mM, green line: $n = 32$ cells from 3 different experiments), combination of 2DG and FCCP (red line: $n = 34$ cells from 3 different experiments) or combination of 2DG and oligomycin (blue line: $n = 33$ cells from 3 different experiments). The stimulus was added at 1 min. (D) Comparison of the amplitudes of responses shown in (C). The amplitude was calculated as a difference between the average of R/R_0 before (0–1 min) and after (9–10 min) stimulus. *: $p < 0.05$ (Tukey's test, Two-sided). Error bars in this figure: S.E.M.

6. line 175, Why were the H₂O₂-elicited responses enhanced by oligomycin or FCCP? Was the binding force between ATP and Mg²⁺ changed by these chemicals?

H₂O₂ decreased cellular ATP concentration as shown Figure 3. In the cytosol, ATP normally binds to Mg²⁺ in the form of an Mg–ATP complex. When ATP is consumed and degraded to ADP, the Mg²⁺ is dissociated because ADP has a lower binding constant for Mg²⁺ than ATP. This leads to an increase in free Mg²⁺. Combination of H₂O₂ with oligomycin or FCCP enhanced the decreased cellular ATP concentration, leading to enhancement of the increase in [Mg²⁺]_{cyto}.

7. Did you check the protective effect of Mg²⁺ against H₂O₂-induced toxicity under your experimental conditions?

We confirmed positive effect of Mg^{2+} on H_2O_2 toxicity as following figure. We added this result in our manuscript as Fig. 5.

Fig. 5 Mg^{2+} supplementation attenuated H_2O_2 -induced cell death.

Viability of the cells exposed to 1 mM H_2O_2 for 24 h in normal culture medium (gray) and culture medium supplemented with 5 mM Mg^{2+} (orange). Bar graphs indicate average and each pair of dots connected line indicate data of each experiment (n = 6). *: $p < 0.05$. (Student's t-test, one-sided).

Reviewers' comments:

Reviewer #1 (Remarks to the Author):

IN this study the authors attempted to visualize the spatio-temporal dynamics of changes in cellular Mg²⁺ content in keratinocytes in response to H₂O₂ exposure. The study follows the notion that ROS decrease mitochondrial membrane potential and that Mg²⁺ protects mitochondria from oxidative stress. Exposure of adult-derived keratinocytes to H₂O₂ inhibited mitochondrial ATP production and resulted in an increase in cellular Mg²⁺, suggestive of a mobilization of Mg⁺ from the Mg-ATP pool. The possibility that this increase in Mg²⁺ occurred as a result of changes in trans-membrane Mg²⁺ fluxes can be excluded based on several controls provided by authors at the request of the previous revision process. The evidence that an increase in Mg²⁺ content limited the decrease in mitochondrial membrane potential induced by H₂O₂ support a possible protective role of Mg²⁺ on MMP.

General comments:

The main comments and criticisms raised at the time of the initial submission were properly addressed in this revision.

Reviewer #2 (Remarks to the Author):

The manuscript was significantly modified but there are still some items that have to be fixed. The main concerns are related to discussion that sounds a bit chaotic. In one part of discussion authors claim In summary, this study revealed that H₂O₂ induced Mg²⁺ dissociation from ATP and that the resulting increase in [Mg²⁺]_{cyto} prevented MMP depolarization in keratinocytes while they claim latter Interestingly, changes in Mg²⁺ appear to precede changes in ATP; this suggests the involvement of another Mg²⁺ source and/or Mg²⁺ transport mechanism in this response. What is true? Is increased Mg results of Mg dissociation from ATP or is it increased because of another mechanism that was not described?

I do not understand meaning of the next sentence: The reduction of [Mg²⁺]_{cyto} in response to FCCP, which is an H⁺ ionophore, suggests that the change in H⁺ electrochemical potential leads to Mg²⁺ transport in keratinocytes.

Conclusion part of discussion is missing.

Reviewer #3 (Remarks to the Author):

no comments

Reviewer #2 (Remarks to the Author):

The manuscript was significantly modified but there are still some items that have to be fixed.

The main concerns are related to discussion that sounds a bit chaotic. In one part of discussion authors claim In summary, this study revealed that H₂O₂ induced Mg²⁺ dissociation from ATP and that the resulting increase in [Mg²⁺]_{cyto} prevented MMP depolarization in keratinocytes while they claim latter Interestingly, changes in Mg²⁺ appear to precede changes in ATP; this suggests the involvement of another Mg²⁺ source and/or Mg²⁺ transport mechanism in this response. What is true? Is increased Mg results of Mg dissociation from ATP or is it increased because of another mechanism that was not described?

Our major finding is that Mg²⁺ dissociation from ATP is essential for Mg²⁺ mobilization in response to H₂O₂ application in keratinocytes. Moreover, we demonstrated that the influx and efflux of Mg²⁺ ion channels and Mg²⁺/Na⁺ transporter and its release from mitochondria are *independent of H₂O₂-induced increase in [Mg²⁺]_{cyto}*: these pathways are major Mg²⁺ sources previously reported in other cell types (Reference No. 14, 16, 19, 22, 29–31, 34–36 in our manuscript). From these data, we concluded that *Mg²⁺ dissociation from ATP is the primary in response to H₂O₂*. However, it is still obscure if dissociation from ATP is the only source of Mg²⁺ in response to H₂O₂ because it is impossible to completely suppress ATP synthesis and consumption under stress condition. We had, therefore, referred the involvement of the other mechanisms in the original manuscript. As you pointed out, our manuscript was difficult to understand for these descriptions in Discussion. So, to make the Discussion section more clear-cut, we removed the sentence referring to the unknown mechanism which might exist and have a minor role in the change in [Mg²⁺]_{cyto} in response to H₂O₂.

I do not understand meaning of the next sentence: The reduction of [Mg²⁺]_{cyto} in response to FCCP, which is an H⁺ ionophore, suggests that the change in H⁺ electrochemical potential leads to Mg²⁺ transport in keratinocytes.

We believe that mitochondria contain some Mg²⁺ also in keratinocytes because enzymes in mitochondrial require Mg²⁺. However, FCCP-induced depolarization of mitochondrial inner-membrane potential did not lead to a Mg²⁺ release from mitochondria in keratinocytes (Fig. 2B) different from neurons. Interestingly, FCCP induced decrease in [Mg²⁺]_{cyto} in keratinocytes, instead. Since FCCP is a H⁺ ionophore, it collapses H⁺ electrochemical gradients across the cell membrane, mitochondrial inner-membrane and other organelle membrane and changes pH in cytosol and organelles. Considering FCCP decreased [Mg²⁺]_{cyto}, changes in H⁺ electrochemical gradients or pH might activate Mg²⁺ transport in keratinocyte. Although this is interesting phenomenon, *this is also independent of H₂O₂-induced change in [Mg²⁺]_{cyto}*. As you mentioned, discussion about FCCP-induced change in [Mg²⁺]_{cyte}, which is not involved in H₂O₂-induced change in [Mg²⁺]_{cyte}, makes the logic difficult to understand. Therefore, we removed the sentence which you pointed out.

Conclusion part of discussion is missing.

We added some sentences summarizing this study and make the conclusion part as a final paragraph.